# Machine-Learning-Based Model for Hurricane Storm Surge Forecasting in the Lower Laguna Madre

Cesar Davila Hernandez [1] , Jungseok Ho [2,*], Dongchul Kim [3] and Abdoul Oubeidillah [2]

1 Department of Civil, Architectural and Environmental Engineering, The University of Texas at Austin, Austin, TX 78705, USA; cesardavilahernandez@utexas.edu
2 Department of Civil Engineering, The University of Texas Rio Grande Valley, Edinburg, TX 78539, USA
3 Department of Computer Science, The University of Texas Rio Grande Valley, Edinburg, TX 78539, USA
* Correspondence: jungseok.ho@utrgv.edu

**Abstract:** During every Atlantic hurricane season, storms represent a constant risk to Texan coastal communities and other communities along the Atlantic coast of the United States. A storm surge refers to the abnormal rise of sea water level due to hurricanes and storms; traditionally, hurricane storm surge predictions are generated using complex numerical models that require high amounts of computing power to be run, which grow proportionally with the extent of the area covered by the model. In this work, a machine-learning-based storm surge forecasting model for the Lower Laguna Madre is implemented. The model considers gridded forecasted weather data on winds and atmospheric pressure over the Gulf of Mexico, as well as previous sea levels obtained from a Laguna Madre ocean circulation numerical model. Using architectures such as Convolutional Neural Networks (CNN) and Long Short-Term Memory (LSTM) combined, the resulting model is capable of identifying upcoming hurricanes and predicting storm surges, as well as normal conditions in several locations along the Lower Laguna Madre. Overall, the model is able to predict storm surge peaks with an average difference of 0.04 m when compared with a numerical model and an average RMSE of 0.08 for normal conditions and 0.09 for storm surge conditions.

**Keywords:** machine learning; storm surge; hurricane; forecasting; CNN; LSTM





## 1. Introduction

The United States mainland has experienced around 280 hurricane strikes since the 1850s. Of these hurricane impacts, nearly a hundred have been classified in the Saffir/Simpson Hurricane Wind Scale (SSHWS) as a category 3 or greater. The monetary damage that such hurricane impacts have can ascend to billions of dollars, as was the case with Hurricane Katrina in 2005 and Ike in 2008. The quantified damages are only a single measure of how destructive a hurricane can be and serve as reminders of the importance of preparation and adequate planning for such events [1,2]. The Laguna Madre, located in South Texas, is one of the six hypersaline lagoons in the world. It is a unique ecological system that provides the perfect environment for the proliferation of numerous species of flora and fauna. This lagoon, and the surrounding region, is impacted by hurricanes that affect the coastal population with flooding and storm surges. Although communities are well aware of the risks that every hurricane season brings, the tools available to prepare and plan are scarce. Storm surge research in this region is paramount to answer the needs of the population. This study seeks to provide a tool that can be used for forecasting storm surge conditions days ahead, without the usage of expensive resources and with automation capabilities. The model proposed here can help first responders and emergency bodies to assemble resources and develop plans ahead of a hurricane impact and subsequential storm surge.

Coastal cities have experienced a boom in growth since the 2000s. The increase has stayed constant at a rate of approximately one percent per year. Leisure has been one of the

most cited reasons for growth, and as such, the need for infrastructure in coastal cities has increased proportionately. There have been projects prompted from coastal growth, such as the construction of transportation, water and electrical infrastructure. This has brought many benefits to coastal communities and has allowed and aided their continuous growth, but at the same time, it has also raised a major weakness point. All the infrastructure necessary to sustain and expand communities in coastal areas are just new vulnerabilities. One of the major drivers of hurricane damages in coastal cities are storm surges, due to their proximity to the ocean. Storm surge refers to the abnormal rise in ocean levels beyond the predicted astronomical tides as a result of sustained winds, among other factors [3]. The state of Texas has many coastal cities that could be potentially struck by hurricanes and subsequent storm surges. Major hurricane impacts can bring destruction to vulnerable infrastructure, creating a potential avenue for billions of dollars worth of damages [4]. It is also important to mention that the danger of a hurricane storm surge is not only limited to direct structural damages; it also represents a worrying environmental risk. Many of the coastal cities that could be potentially damaged by hurricanes and storm surges also house ports. These ports expose industrial complexes to catastrophic events. As an example, Hurricane Ike brought USD 30 billion dollars in damages to the cities of Houston and Galveston, where at least 112 deaths occurred. The Houston Ship Channel is one of the busiest seaports in the world and is the host of many petrochemical complexes, which heightens the potential for an environmental disaster [5,6].

Since storm surges have the potential to cause damages worth billions of dollars, as well as cause deaths and possible environmental disasters, it is of the utmost importance to plan accordingly when a major hurricane is approaching a coastal area. It is possible to assess the potential risk that a hurricane poses in terms of storm surge by creating a simulation of the interaction between winds, atmospheric pressure, tides, and waves. To date, the problem of simulating hurricane storm surges has been solved through the usage of computer models capable of capturing these interactions and producing fairly accurate storm surge estimates. Some of the computer models that are available and are currently being used to predict storm surges by agencies such as the Federal Emergency Management Agency (FEMA) or the US Army Corps of Engineers (USACE) are Advanced CIRCulation (ADCIRC), or the Sea, Lake, and Overland Surges from Hurricanes (SLOSH) model [7–9]. There exist other numerical models for different purposes, such as TxBLEND, developed by the Texas Water Development Board (TWDB) used to estimate salinity conditions for Texas estuaries [10]. The estimation of storm surges is not only a matter of accuracy; it is also a problem of time. Emergency preparations are time-sensitive; numerical-based storm surge models such as ADCIRC or SLOSH require a lot of time to be executed, especially if there are not many resources available. Currently, high-performance computer clusters (HPC) are employed to run such models on a large scale and provide enough resources for their computations to be timely. It is important to mention that such models are often coupled with wave models, which add another level of complexity, raising the resource requirements of the models. Some examples of the wave models used in conjunction are the Wave prediction Model (WAM), the Steady-State Spectral Wave Model (STWAVE), and Simulating WAves Nearshore (SWAN) [7,11]. Models such as ADCIRC run their computations based on an unstructured mesh containing bathymetry information of the area to be simulated; this mesh is a discretization of the area that needs sufficient detail near points of interest to better capture the physics involved. It is because of this that there is always a tradeoff between mesh resolution and the time required to complete computations. The ADCIRC code is optimized to scale and parallelize very efficiently, but if, as mentioned, there is a coupled model meant to simulate waves, then the complexity of the model scales vastly, which can hog the computational resources available. It is easy to see how high-fidelity models such as ADCIRC are out of the reach of endeavors without substantial funding, and the long runtime and high costs represent a limiting factor for timely emergency notifications if resources are lacking. The prohibitive costs of a numerical model and the demand for timely storm surge emergency notifications

pushed the search for a way to develop an Artificial Intelligence (AI) model for accurately predicting storm surges without the need for large amounts of computational resources. In this study, a machine-learning-based storm surge forecasting model is proposed and created for predicting storm surges at discrete points along the Laguna Madre in Texas. The goal of the study is to create a machine learning model capable of predicting storm surges by using only a fraction of the computational resources that numerical models use.

## 2. Materials and Methods

To create this study, a literature review was conducted first to gather information on what previous attempts have been made to create machine-learning-based models for storm surge prediction. The literature review shed a light on what types of models were used as well as what predictors are employed.

### Literature Review

Machine learning techniques have been employed extensively in the prediction of weather and for the modeling of complex relationships, such as storm surges, precipitation, and floods. Machine learning has proven itself to be a valuable tool in the creation of very accurate, non-resource-intensive models that can capture very complex phenomena. For example, Artificial Neural Networks (ANN) have been utilized for capturing the rainfall–runoff relationship in basins where the declaration of the internal structure of the watershed is not needed [12]. Neural networks have also been used to predict floods with fairly good accuracy [13]. These initially reviewed papers reiterate the possibility of creating a machine learning model for storm surges. Hurricane storm surges are an example of a complex nonlinear relationship where the usage of machine learning methods can prove to be very beneficial. Neural networks are a type of machine learning technique that have already been proven successful for storm surge prediction. In the past, several studies have explored the performance of neural network architectures when it comes to storm surge estimation. Simple ANNs [12,14–21] have already succeeded in recognizing the relationship between weather variables and the subsequent storm surge; however, some problems still remain. For a better visualization, Table 1 contains a sample of 10 reviewed studies.

As it can be seen from Table 1, most of the studies utilized ANN to produce their storm surge predictions. ANNs accept a fixed amount of predictor variables; the most common predictor variables utilized in the studies reviewed are storm parameters. Some of the parameters are the location of the storm, angle of approach, translation speed of the storm, wind speeds, and radius of strong winds. The accuracies obtained by the studies are good; however, there are limitations that could be improved upon.

**Table 1.** Sample of papers reviewed. Predictors, data types, and metrics used in each paper can be easily referenced in the table.

| Paper | Predicted | Model | Predictors | Data Type | Metrics |
|-------|-----------|-------|------------|-----------|---------|
| [16] | Storm surge | ANN | Longitude, latitude at landfall, heading direction, central pressure, moving speed, maximum wind speed, radius of the strong wind speeds | 59 Historical storms | CC |

**Table 1.** *Cont.*

| Paper | Predicted | Model | Predictors | Data Type | Metrics |
|-------|-----------|-------|------------|-----------|---------|
| [17] | Normalized storm surge | ANN | Pressure, wind velocity, wind direction, estimated astronomical tide | Historical storm descriptive parameters | RMSE and CC |
| [20] | Storm surge | RBF, GRNN, MLP3, MLP4 | Two experiments: (1) daily mean sea level from preceding day, 6 h forecast of wind speed, direction. (2) 4–10 different parameters | Historical storms from 1950–1999 | RMSE and CC |
| [22] | Max still-water inundation, runup, wave height | Stats. | Landfall location, angle at landfall, central pressure, forward speed, radius of maximum winds | 1500 synthetic storms | MSE |
| [15] | Storm surge | ANN | Atmospheric pressure and winds | Historical NCEP-NCAR data | RMSE and CC |
| [12] | Tide, storm surge | ANN, ANFIS | Wind speed, wind direction, air pressure, simulated water level using hydrodynamic model | Historical data | MAE, RMSE and PE |
| [19] | Storm tide, coastal inundation | ANN | Landfall location, approach angle, translation speed, wind speed | Computed storm tide, coastal inundation by ADCIRC | CC |
| [14] | Storm surge | ANN, GPR, SVR | Storm parameters, reference latitude and longitude of storm as well as coastal points | USACE NACCS synthetic data | MSE, RMSE, CC |
| [21] | Tidal level | SVR, ANN, CNN, LSTM | Previous and current tidal water level | 21 years of historical data from tide stations | RMSE, MAPE |

The study by [23] developed a multioutput artificial neural network model which was used to predict storm surges in the North Carolinian coast. The authors mention a couple limitations of ANNs; for example, they found out that ANNs often underestimate peak surges. Furthermore, they concluded that the underestimations could be a result of the memoryless approach of ANNs. Naturally, including memory in an ANN-based machine learning model could help improve the results. The usage of memory in neural networks for tidal prediction was explored by [21], where they compared many approaches for predicting ocean water levels at 17 different stations in Taiwan. The model utilized a type of neural network called Long-short Term Memory (LSTM). LSTMs provide a solution to the memoryless problem mentioned by [23] and outperformed other methods. The results of LSTM show their potential for usage in storm surge prediction. It is important to mention that the study in [21] focused only on tidal levels, and no storm surge or weather conditions were considered.

LSTMs are an example of Recurrent Neural Networks (RNNs). These types of neural networks are often used for process control or time series predictions [24]. LSTMs can solve one of the glaring problems that are found in most studies referenced in Table 1, where the usage of ANNs limited the performance of the resulting models due to their lack of

'memory'. Improvement on time series prediction is expected with the usage of LSTMs in comparison with ANNs, and that is why they were chosen for further exploration in this study in comparison with regular ANNs.

Convolutional Neural Networks (CNNs) are another type of neural network architecture that are utilized in this study. CNNs are a very common and well-known neural network architecture. Their structure, often comprised of convolutional, pooling, and fully connected layers, has driven forward the field of computer vision in the past decades. Modern iterations of CNNs were first introduced by [25]. AlexNet [26] brought a scaled-up version of CNNs with around 60 million parameters. Since then, the usage of CNNs has skyrocketed, and their applications in several disciplines have been popularized. In this study, CNNs are used as part of the model to read and interpret weather information obtained in gridded form.

## 3. Modeling Scenarios and Data Processing

The storm surge forecasting model developed requires two sources of data. The first is a database of ADCIRC numerical model predictions. The second source of data corresponds to forecasted gridded weather data.

### 3.1. ADCIRC Numerical Model Database

The details of the ADCIRC model utilized to create the database of results for the Laguna Madre can be seen in the following study [27]. To execute ADCIRC and create the dataset necessary for training the machine learning model, it is necessary to develop two files for each simulation. The geometric properties of the model, as well as the nodal parameters, remained constant for all simulations. One of the two files utilized is named Model Parameter and Periodic Boundary Condition file, or 'fort.15' [28]. The 'fort.15' file is used to set the parameters that configure the ADCIRC model for running. The date of the simulation, duration, and tidal constituents are just some relevant parameters that were changed as required, while other parameters remained constant for all simulations, such as the type of input file used.

### 3.2. Forecasted Gridded Weather Data

The second and perhaps more critical file is 'fort.22', or Meteorological Forcing Data [28]. The machine learning model and the ADCIRC numerical model utilize the same input source: a gridded forecasted weather dataset named the North American Mesoscale Forecast System (NAM) [29]. The NAM dataset provides continuous forecasted gridded atmospheric conditions over the continental United States. The model is distributed by the National Centers for Environmental Information (NCEI) and consists of a 12 km-resolution grid with a forecasting range of up to 84 h. This forecasted gridded dataset is utilized in two forms. First, a set of input files for the ADCIRC numerical model is created; at the same time, the dataset is converted into image files to serve as input for the machine learning storm surge model.

To create both ADCIRC's and the machine learning storm surge model input, the NAM dataset is trimmed to the domain of interest; in this case, the Gulf of Mexico. After the NAM data are trimmed spatially, only the variables that will be used are kept, which includes the U and V components of winds at 10 m elevation and atmospheric pressure. The data is then projected into a regularly spaced grid over the Gulf of Mexico. The resulting grid can be directly used for input in ADCIRC and is also leveraged for the creation of images for their use in the machine learning model. To create the input images, each variable in the data is normalized and its values mapped into the range of an unsigned byte, from 0 to 255, for the creation of PNG stills with three color channels. The resulting composite can be seen below in Figure 1.

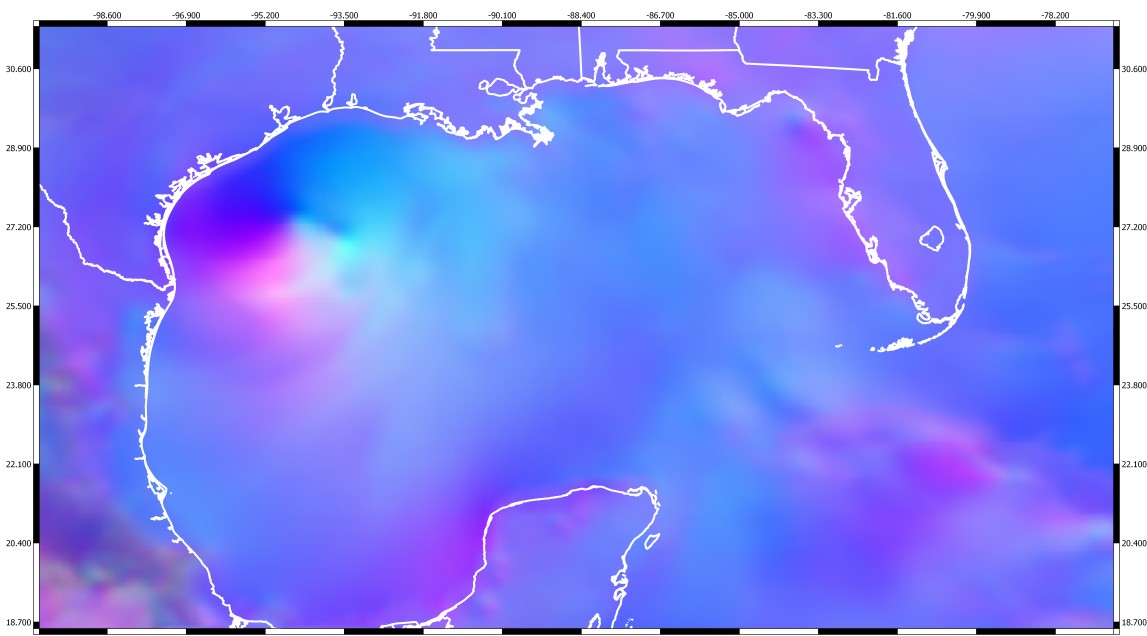

**Figure 1.** Image showing hurricane Hanna (2020) over the Gulf of Mexico as it is approaching the coast of Texas on the morning of July 25th. The image was constructed from retrieved NAM forecasts, where the color channels represent U and V components of winds and atmospheric pressure.

To test the machine learning storm surge forecasting model, a total of five scenarios were prepared from the available data. These five scenarios were selected as representative of the presence and lack of storm surge conditions. Three of the five scenarios are representative of hurricanes that impacted the Laguna Madre directly or caused fluctuations in the ocean levels in the Laguna Madre; the remaining two scenarios provide everyday or normal conditions, meaning no major weather event occurred near the Laguna Madre.

### 3.2.1. Hurricane Dolly (2008)

Dolly made landfall as a category 1 hurricane on the Saffir–Simpson Hurricane Wind Scale at South Padre Island, Texas, with estimated maximum winds of 86 mph. The storm reached peak intensity at around 1400 UTC on 23 July, 4 h before landfall, centered less than 20 nautical miles east of the Rio Grande River. Part of Hurricane Dolly's track can be seen in Figure 2 below.

### 3.2.2. Hurricane Alex (2010)

Alex made landfall as a category 2 hurricane on the Saffir–Simpson Hurricane Wind Scale near Soto la Marina, Tamaulipas in northeastern Mexico. At landfall, Alex had an estimated maximum wind speed of 109 mph at around 0200 UTC on 1 July. The path followed by Hurricane Alex as it made landfall can be seen on Figure 2 below.

### 3.2.3. Hurricane Hanna (2020)

Hanna made landfall as a category 1 hurricane at Padre Island, Texas. The hurricane reached a peak intensity of 92 mph as it was located off the coast of South Texas at 1800 UTC on 25 July. Hanna weakened to a tropical storm by 0600 UTC on 26 July and dissipated at 1800 UTC on 26 July as it neared Monterrey, Mexico. The path Hurricane Hanna took can be seen in Figure 2.

### 3.2.4. Normal Conditions: June 2008

During this month, precipitation for the southern region of the United States was below normal, with some regions receiving lower than 5% of the average June rainfall. A

single tropical storm named Arthur formed on May 30th near the shore of Belize and, after two days, dissipated over the Yucatan Peninsula in Mexico.

### 3.2.5. Normal Conditions: June 2020

This month was especially dry with a precipitation total for the contiguous U.S. of about 0.21 inches below average. Two tropical storms were recorded in the Atlantic, Amanda and Cristobal. Amanda made landfall in Guatemala and its remnants developed to form Cristobal, which eventually made landfall in Louisiana, just east of Grand Isle.

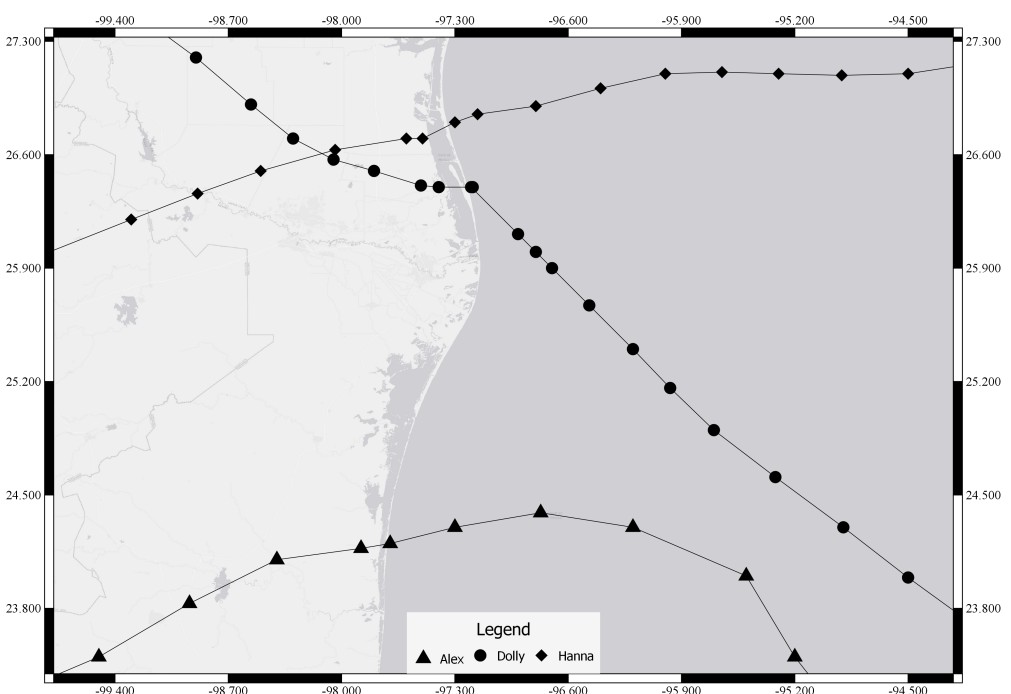

**Figure 2.** Paths followed by Hurricane Alex, Dolly, and Hanna as they made landfall in Texas. The white line depicts the U.S.–Mexico border.

It is important to note that while data are abundant, a bias was identified, and its impact on the performance is later discussed at the end of the paper. The root of the bias comes from the data available in the area where the study was conducted. Data from NAM provided a total of 13 hurricane seasons for training. Each season translates to six months of data, meaning a total of more than 6 years of continuous data was available. During this period of time, the Lower Laguna Madre saw the impact of around 20 abnormal sea-level conditions due to weather events. On average, the duration of such abnormal events was one week. This translates to 6% of the data being representative of storm surge conditions. This is problematic, since it evidences a bias in the data towards normal conditions; however, this was expected since disastrous impacts of hurricanes are scarce on a local level in the South Texas region.

## 4. Forecasting Model

There are three forecasting models that were created during this study. The differences between them relies mainly on the machine learning architectures used and their coupling. The two types of architectures used in the models created are CNN, LSTM and CNN+LSTM.

As explained before, CNNs are a type of artificial neural network architecture specialized in the analysis of image data. CNNs are inspired in the biology of the visual cortex of animals. They are great at extracting relevant features out of images and, given that the type of data used for the realization of this study can be interpreted as an image, the usage of CNNs suits the type of problem at hand. The first model created only considers image data as its input and produces a time series of ocean elevations corresponding to a specific

time interval in the future. There are a total of 4 CNN-LSTM layers, with pooling operations in between them that accept the time series of images of future weather conditions. Their output is then fed to a series of regular densely connected neural network layers where the final time series output prediction is generated. Training parameters are the same across the CNN and LSTM models: 50 epochs, 6 samples per batch, and a validation split of 25%.

The second model to be evaluated is based on the LSTM neural network architecture. It is composed of 50 LSTM units that connect to a series of regular neural network layers where the output is generated. The training parameters are the same as the CNN model above. The model will not utilize image data for its input; instead, it will use time series data corresponding to past conditions of water surface elevation. Using this information, the model will create a prediction of future conditions.

The third model is a combination of both architectures and can be thought of as the final forecasting model. The two previously described models are simply a set of preliminary attempts to judge the capabilities of the architectures to establish a connection between past ocean surface elevations and weather and future ocean surface elevations. The final surrogate model combines both architectures to take more information into account for the generation of storm surge predictions. The model can be classified as a mixed-input model with two heads. The first head corresponds to the CNN model, using the same architecture discussed in the CNN-only model and accepting the same time series input of images. The second head of the model is the LSTM-only model, with the same architecture and input of time series elevations. Their outputs are then concatenated and fed to a series of densely connected neural network layers that produce the final output. This final model was trained on 100 epochs with a smaller batch size of 3 samples to accommodate the size of the model in the GPU. All three models were trained on the same hardware, an RTX 3060 NVIDIA GPU with 12 GB of VRAM. The training time for the CNN model took around an hour, while the LSTM model only took 5 min to train; however, the CNN+LSTM mixed-input model's total training time was around 7 h. Mixed precision was also leveraged to achieve speed ups.

To train the respective models, two sets of data were utilized. First, a set to train the preliminary models to evaluate their performance based on data from a recording station. Second, a set to train the final surrogate model based on synthetic data generated with the ocean circulation numerical model. Both sets of data have corresponding water surface elevation and forecasted weather conditions data. The first, or preliminary training dataset, contains water surface elevation data coming from the Center for Operational Oceanographic Products and Services (CO-OPS) Port Isabel recording station in Texas, with ID: 8779770. The data obtained from the Port Isabel recording station has the same coverage as the forecasted weather information.

The data obtained from the Port Isabel recording station was leveraged as part of the initial investigation into the feasibility of predicting storm surges with the LSTM and CNN architectures both separately and combined. As part of a pilot, the Port Isabel recordings were used to iterate models by tuning the hyperparameters and their architecture to find the best performance. The best-performing architecture obtained from the Port Isabel pilot models was used in the first iterations of the final model trained on the synthetic water surface elevation data. Subsequent iterations changed the size of the model, integrated mixed precision, and modified training parameters to find the best performance for the final CNN+LSTM model.

The second set of data used to train the final storm surge model comes from the execution of numerical model simulations corresponding to each Atlantic hurricane season from 2008 to 2020. To force the simulations, the forecasted weather information was used. This provides a training dataset for the surrogate model to learn and replicate the performance of the numerical simulations. To evaluate the machine learning model, a set of 10 virtual buoy stations were selected from the numerical model finite element mesh. These virtual stations correspond to points of interest in the Lower Laguna Madre. The set of virtual buoy stations and their location can be seen in Table 2.

**Table 2.** Table detailing the locations of virtual buoy stations used in the study.

| Name | Latitude | Longitude | Numerical Model Mesh ID |
|---|---|---|---|
| South Padre Island (SPI) | 26.0854 | −97.1562 | 42195 |
| Laguna–SPI | 26.0862 | −97.2007 | 59162 |
| Ship Channel | 26.0423 | −97.2071 | 61388 |
| Laguna Heights | 26.0854 | −97.2518 | 55382 |
| Laguna Vista | 26.1007 | −97.2815 | 52698 |
| Port Isabel | 26.062 | −97.215 | 60551 |
| Port Mansfield Inside | 26.5588 | −97.4201 | 5684 |
| Port Mansfield Outside | 26.564 | −97.2593 | 10629 |
| Arroyo Colorado Inside | 26.3616 | −97.3266 | 29783 |
| Arroyo Colorado Outside | 26.3814 | −97.1979 | 18356 |

To better visualize the location of the stations, a map of the Lower Laguna Madre with the virtual stations can be seen below in Figure 3.

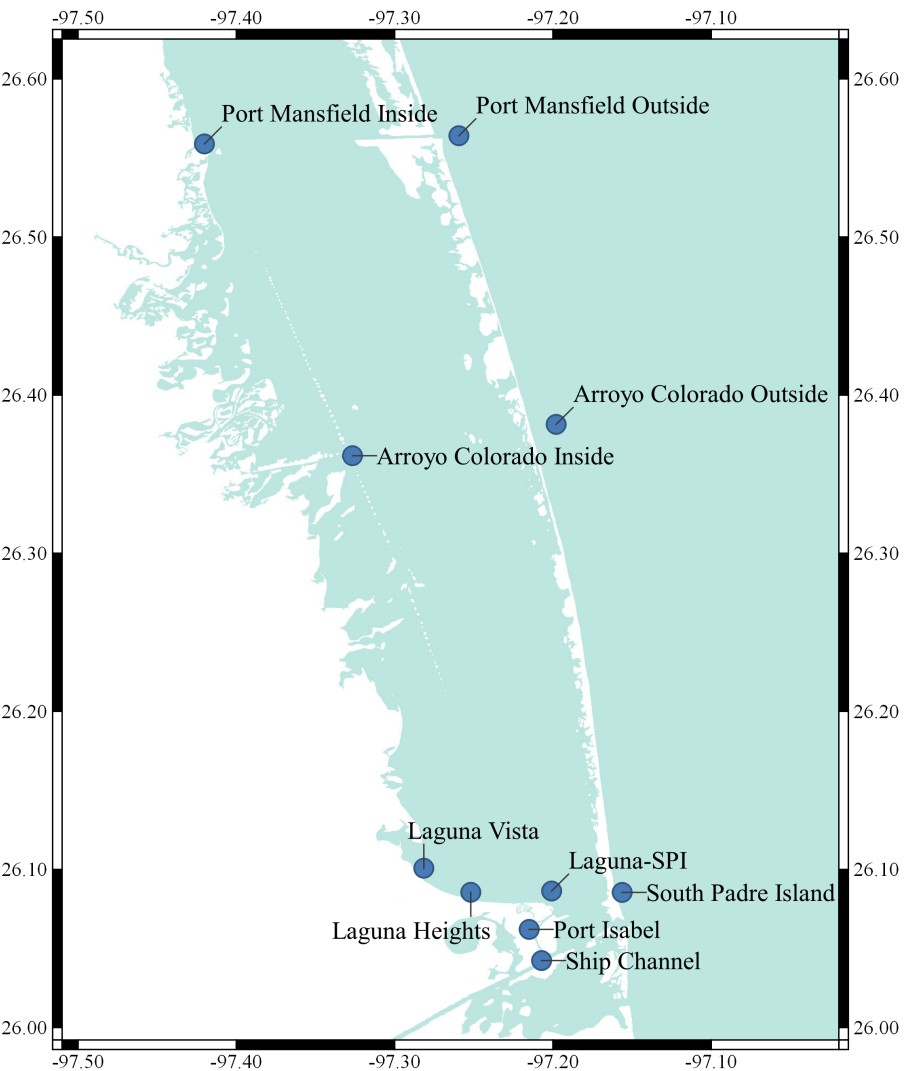

**Figure 3.** Map for the location of the 10 virtual buoy stations in the study.

## 5. Results

Two different sets of results are presented. First is the preliminary set of results which corresponds to three different models trained on data recorded by the single Port Isabel station. The first model only utilizes weather information and CNNs to preform storm surge prediction; it is expected that these results are far from accurate since there is no input with relevant information regarding tide harmonics, only weather conditions. The second preliminary model takes into consideration only previous water surface elevations to perform predictions without the influence of weather information using only LSTM. This model was expected to outperform the CNN model in at least normal conditions, since having information about previous tide elevations is sometimes sufficient to predict future conditions. The third preliminary model corresponds to a CNN+LSTM coupled model with mixed input. The third model accepts future weather information in the form of images, as well as past surface elevations as time series data.

The second set of results corresponds to the final machine learning model created for each of the 10 virtual buoy stations with training and validation data generated by the numerical model simulations.

To evaluate the performance of the models, the set of scenarios previously discussed in Section 3 was used.

### 5.1. Preliminary Modeling Results

These results help to illustrate the influence that the CNN and LSTM architectures have on the final model. The Root Mean Squared Error (RMSE) metric was used to compare their performance.

### 5.1.1. CNN-Only Model

As previously discussed, the CNN model is expected to be the weakest of the three. This model only takes into consideration weather information, completely ignoring previous water surface elevations. In the case of June 2008, where no major hurricane occurred, the results exhibit a pattern that does not follow water surface elevations, as seen in Figure 4. This period of time was chosen to better illustrate the importance of considering tide harmonics in the model.

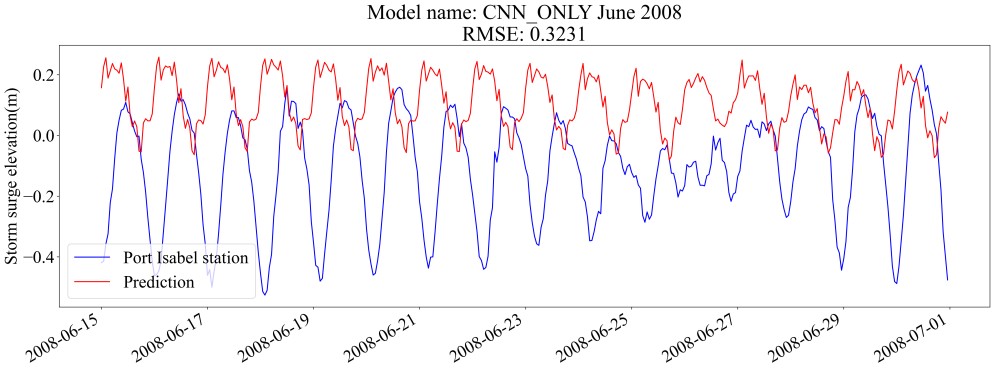

**Figure 4.** Preliminary CNN-only model prediction for the month of June 2008.

The CNN model does not encounter fluctuations during this month that could hint at a storm surge event. It defaults to an oscillation pattern to maximize its score. The inability of the model to predict the elevation might be due to its lack of knowledge of ocean elevations. It is working, in this case, as a detector of storm surge weather conditions rather than a storm surge predictor. The ability of the model to detect storm surge triggering conditions is reflected in Figure 5.

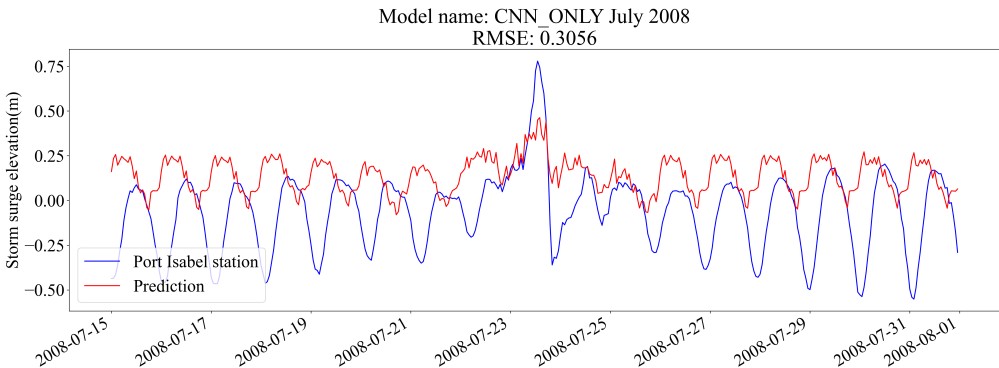

**Figure 5.** Preliminary CNN-only model prediction for the month of July 2008.

Here, the model defaults to a pattern which is disrupted by the detection of a storm surge triggering weather event. While the estimation is completely inaccurate, the detection of a storm is reflected as a change in the pattern, which hints at the ability of the model to detect storms in image data.

### 5.1.2. LSTM-Only Model

The LSTM preliminary model only considers past water surface elevations for its prediction and ignores weather information. The month of June 2010 saw the impact of Hurricane Alex in Mexico. The extent of the storm affected the water surface elevation of the Laguna Madre, which can be seen in the peak of the blue line in Figure 6.

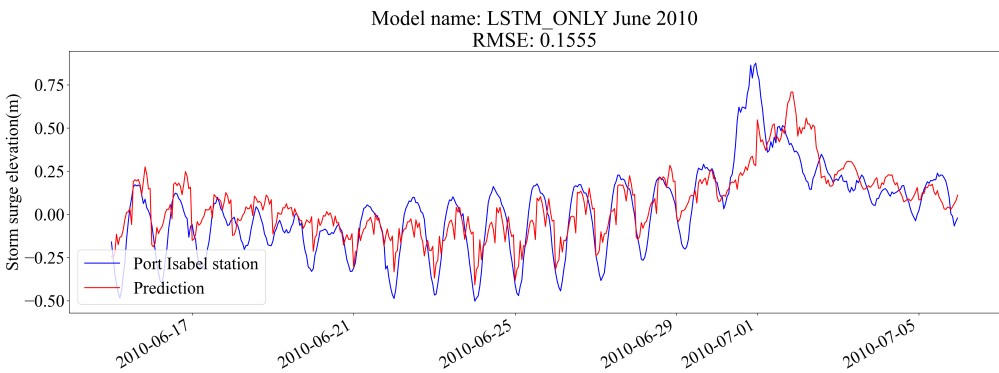

**Figure 6.** Preliminary LSTM-only model prediction for the month of June 2010.

It can be seen from the prediction that when the storm hits, the model tries to replicate what happened but is delayed in its prediction. This is because the model only considers past elevations to construct its prediction, and since storm surges are produced by weather fluctuations, the model has no information to anticipate the surge. Subsequently, the model tries to continue the surge but cannot estimate it accurately.

Even in storm surge conditions, the LSTM architecture outperforms the CNN model in its overall score but fails to detect the storm as it hits and can only produce a delayed response.

### 5.1.3. CNN+LSTM Model

This model is expected to perform better than both individual models since it is considering critical information on weather and previous water surface elevations to produce a prediction. The month of June 2008, when there were no storms hitting the Laguna Madre area, can be seen in Figure 7.

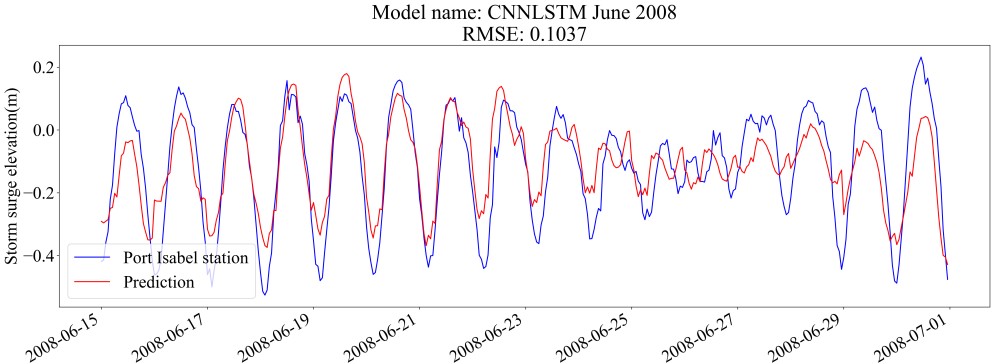

**Figure 7.** Preliminary CNN+LSTM model prediction for the month of June 2008.

It can be seen from this plot that the model is able to follow the trends closely; however, it still struggles in some places by underestimating both highs and lows. The CNN+LSTM model prediction for the month of June 2010 when the impact of Hurricane Alex was felt in the Laguna Madre is a very good example of the performance of the machine learning model, as shown in Figure 8.

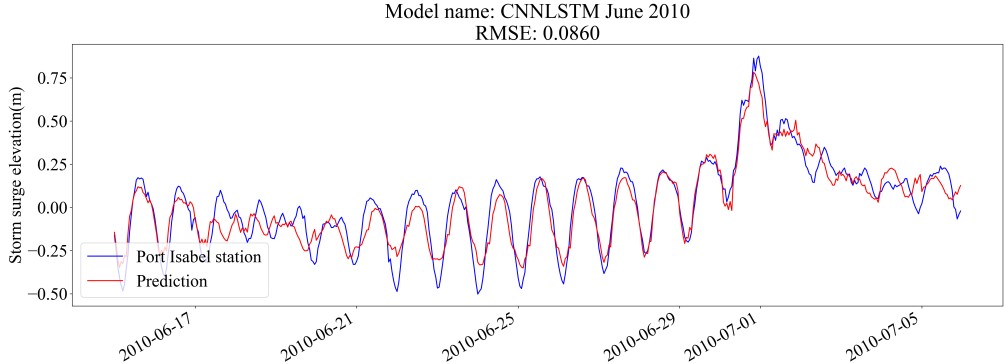

**Figure 8.** Preliminary CNN+LSTM model prediction for the month of June 2010.

In this case, the model is able to follow the surge correctly but still struggles at some points where normal conditions were expected. However, as it can be seen in the figures, the RMSE was reduced from a maximum of 0.1555 in the LSTM-only model in the month of June 2010 (when Hurricane Alex hit) to 0.0860 during the same period of time in the preliminary model that combines CNN and LSTM.

### 5.2. Machine Learning Storm Surge Forecasting Model Results

This set of results was produced by the finalized model trained on the full set of virtual buoy stations. For each of the five testing scenarios, a sample of four buoy stations is presented as a scatter plot. The four buoy stations selected represent important socioeconomic areas in the Laguna Madre.

The results are presented as two separated groups: a group that only considers normal conditions and a second group that includes storm surge conditions.

### 5.2.1. Normal Conditions

The predictions of the final CNN+LSTM model during normal conditions in June 2008 can be seen in Figure 9.

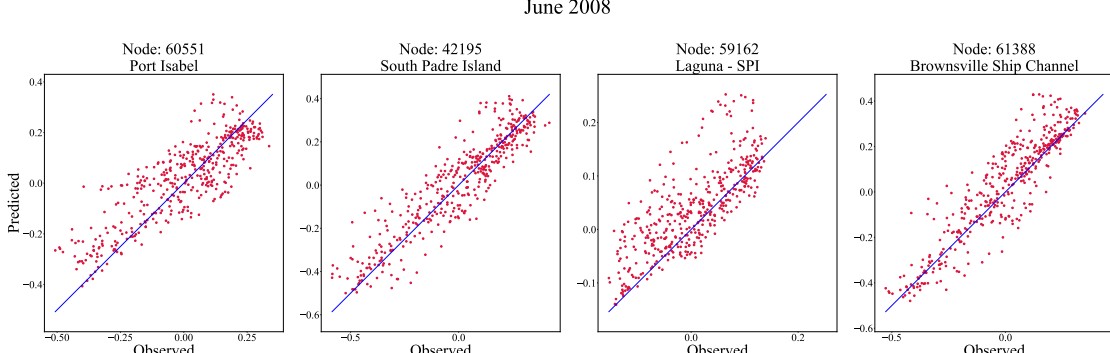

**Figure 9.** Scatter plot for four nodes in the CNN+LSTM final machine learning model for the month of June 2008.

The RMSEs for the four nodes, Port Isabel, South Padre Island, Laguna-SPI, and Brownsville Ship Channel, during the month of June 2008 are 0.1058, 0.0980, 0.0634, and 0.0990, respectively.

In a similar manner, the RMSE for each virtual buoy station for normal conditions in the month of June 2020 is 0.0730, 0.0832, 0.0541, and 0.0743. The scatter plot can be seen in Figure 10 below.

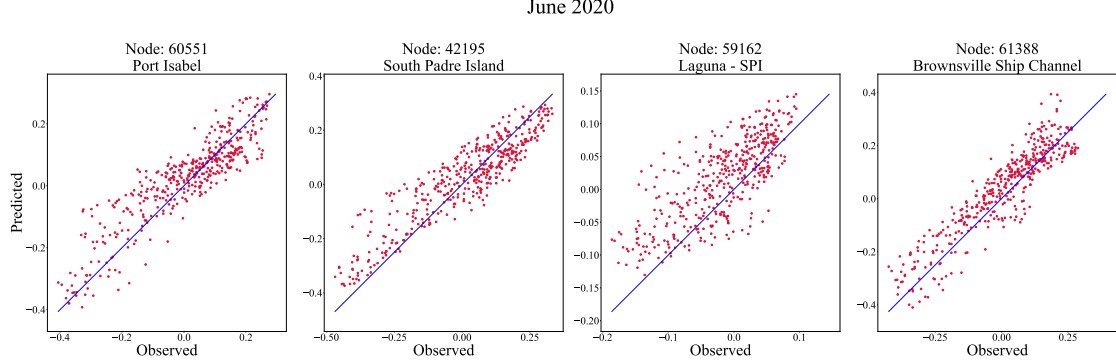

**Figure 10.** Scatter plot for four nodes in the CNN+LSTM final machine learning model for the month of June 2020.

### 5.2.2. Storm Surge Conditions

Hurricane Dolly predictions can be seen in Figure 11. The RMSEs for the different stations during Hurricane Dolly are 0.1312, 0.1632, 0.0462, and 0.1430 for nodes 60551, 42195, 59162, and 61388, respectively. There is more spread in these predictions; however, the peak surges observed and predicted do not differ greatly. On average, there is a 0.1183 m difference between the peak observed and the one predicted.

In the case of Hurricane Alex in 2010, the RMSEs for the four stations that can be seen in Figure 12 are 0.065, 0.0822, 0.0549, and 0.075, respectively. The model performed much better during this event than for Hurricane Dolly. The average difference between the observed and predicted peaks is much smaller compared with that of Dolly, being 0.0422 m.

Finally, for Hurricane Hanna in 2020, the RMSE for each of the buoy stations in Figure 13 is 0.1142, 0.1105, 0.0835, and 0.1268, respectively. Interestingly, the peak observed and predicted average difference is −0.0337 m, meaning, the model tended to overestimate storm surge peaks.

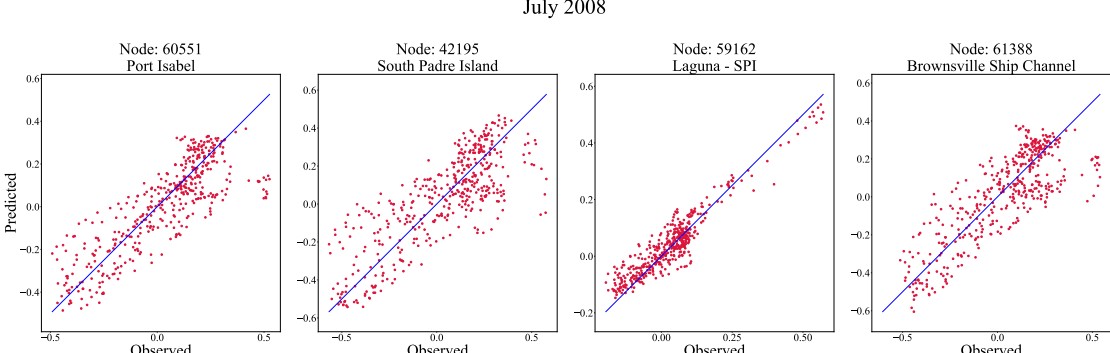

**Figure 11.** Scatter plot for four nodes in the CNN+LSTM final machine learning model for the month of July 2008.

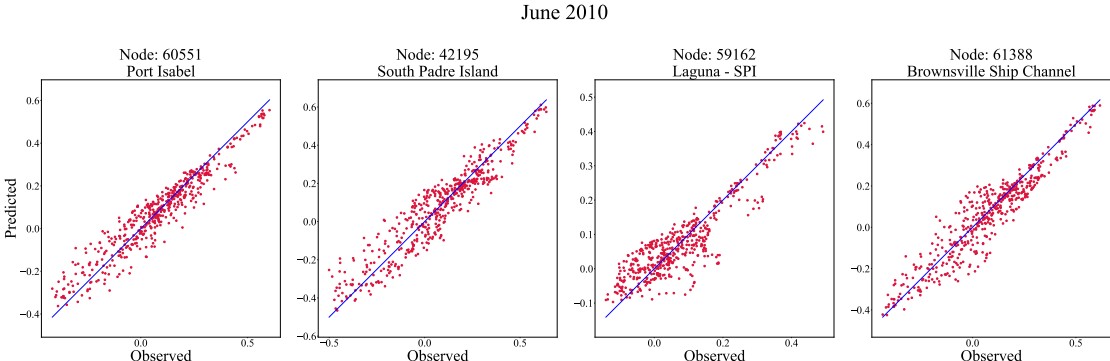

**Figure 12.** Scatter plot for four nodes in the CNN+LSTM final machine learning model for the month of June 2010.

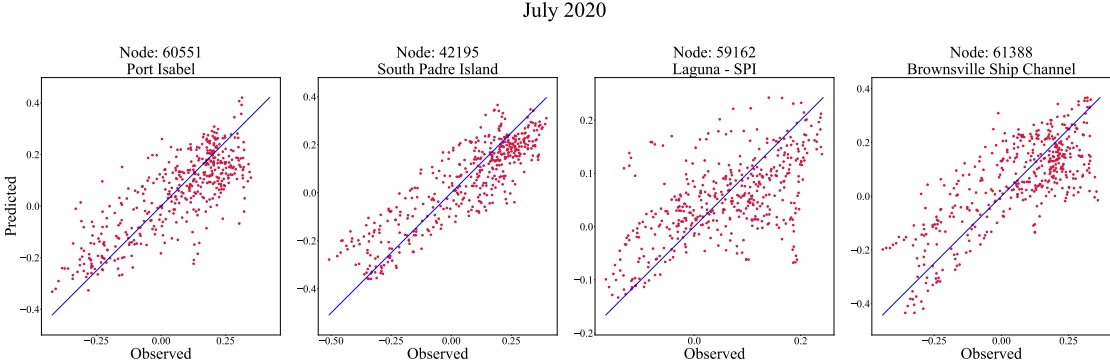

**Figure 13.** Scatter plot for four nodes in the CNN+LSTM final machine learning model for the month of July 2020.

For most cases, the model was able to produce better estimations for everyday conditions in comparison with storm surge conditions. Limitations in data availability contributed to these results. Data representative of normal conditions constitute nearly 94% of the samples, with the rest corresponding to storm surge conditions. The limitation in data is attributed to the area that the study was conducted in, where, since 2008, around 20 abnormal ocean elevations due to weather were recorded. To improve results without modifying the model, the amount of samples representative of storm surge conditions must be increased. To accomplish this, in the field of image recognition, there are a couple of techniques that allow for the creation of more samples for training. There is no use for these algorithms for the perturbation of existing gridded weather forecasts, since the information

contained in the samples is the result of complex atmospheric phenomena, and without a numerical model it is not feasible to perturb the information contained in the samples with traditional image data augmentation algorithms. Nonetheless, the generation of new samples for the improvement of the described machine learning storm surge model is out of the scope of this study. Other possible improvements to the model could come from the input considered which, in essence, is the same type of input used in the numerical model but transformed into a different format. Perhaps the application of feature engineering techniques to the features in the input data could yield better results and a different model architecture that considers a new type of input. Another possible route could be creating an ensemble of models whose output is averaged, which could be achievable without changing the underlying architecture of the model or the input.

## 6. Conclusions

Throughout the set of experiments that were set up for testing the performance of the model, a very noticeable pattern emerged. The machine learning storm surge forecasting model created in this study was able to constantly provide an accurate estimation of everyday conditions with an average RMSE of 0.0813 among the buoy stations presented. Let us recall that these conditions refer to the absence of any impactful weather event that could change the water surface elevations from the expected astronomical tide harmonics. That is, the model was able to capture the harmonic future oscillations from previous observations while considering normal conditions of winds and pressure. This accomplishment is mainly due to the presence of the LSTM architecture. Ref. [21] demonstrated that LSTM networks are capable of predicting tidal levels effectively with superior performance when compared with other methodologies. Being able to correctly predict tidal levels is imperative to predicting storm surges, since the oscillation of tides have an additive or subtracting effect on the final elevations, as it was noted in Section 5.1. Once this baseline was accomplished during the modeling phase, it was important to build upon it. The next logical step was to attempt to incorporate weather information into the model. The inclusion of the CNN architecture improved everyday conditions and allowed for the estimation of storm surge conditions due to the consideration of input from weather events.

On a more specific note, in the preliminary model results presented in Section 5.1.3, a view of the overall performance of the final storm surge model and the importance of the inclusion of both CNN and LSTM architectures can be seen. The result for normal conditions in the Port Isabel recording station is greatly improved in the CNN+LSTM preliminary model from Figure 7 when compared directly with the CNN model prediction from Figure 4, highlighting the importance of the inclusion of the LSTM architecture, as mentioned. There was a reduction in RMSE from 0.3231 for the CNN-only model to 0.1037 for the CNN+LSTM model. The contribution of the CNN architecture to the final model is also important. Figure 6 shows a delayed response of the predicted storm surge in the presence of a storm event and an RMSE of 0.1555. For storm surge conditions, the inclusion of forecasted weather gridded data as input for the CNN head of the model provides a considerable improvement not only in peak prediction but in its timing, as it is seen in Figure 8. This change further reduced the RMSE to 0.0860 during the same period of predicted time.

In Section 5.2, the prediction of normal conditions and storm surge conditions for the final model shows an important point. Overall, normal condition predictions have a maximum RMSE of 0.1058 and an average RMSE of 0.0813. In the case of storm surge conditions, the maximum RMSE is 0.1632 and the average is 0.0996. From this, we can note that the predictions for storm surge conditions are worse than for normal conditions in the final model. This is important, as it shows a limitation in the storm surge forecasting model presented. However, this limitation in accuracy can be directly attributed to the lack of storm surge data training samples in comparison with normal conditions. The objective of the model presented was to be a storm surge forecasting tool, as such, forecasting-type data were used to train it. These data were obtained from NAM [29] forecasts on winds

and atmospheric pressure. This limited the amount of data available, and while there are big collections of parametric data, such as HURDAT2 [30], they were not used in the study because the data are generated after an event and not as a forecast, which could not be directly leveraged in this storm surge forecasting model. Another constraint in data comes from the focus on the Lower Laguna Madre area. A greater model domain will provide better data and plentiful storm events to improve the performance of the model. The logical step is to include a bigger domain and hence more data, which could drive performance further without changing the underlying model architecture. Another approach is data augmentation, which would entail generating synthetic samples of storms.

Overall, the model provided a reliable estimation of storm surge peaks, with the average error between observed and predicted peaks being in the realm of centimeters. The model also tended to underestimate the storm surge peaks, except for the case of Hurricane Dolly, where the model overshot the observed peak by about 0.033 cm on average. Considering the data limitations in this study, a machine learning approach to storm surge prediction using forecasted weather gridded datasets with the inclusion of CNNs is a viable approach. As data continue to be collected and generated, the performance of CNN-based models will continue to improve. This will allow for the possibility of deploying forecasted weather image-based storm surge forecasting solutions that utilize small computational resources during prediction.

**Author Contributions:** Conceptualization, C.D.H., J.H. and D.K.; methodology, C.D.H.; software, C.D.H.; validation, C.D.H.; investigation, C.D.H.; data curation, C.D.H.; writing—original draft, C.D.H.; writing—review & editing, C.D.H. and J.H.; visualization, C.D.H.; supervision, J.H., D.K. and A.O.; project administration, J.H.; funding acquisition, J.H. All authors have read and agreed to the published version of the manuscript.

**Funding:** This project was partially funded by the Texas Coastal Management Program, Texas General Land Office Award Number NA18NOS4190153.

**Data Availability Statement:** The data is not publicly available due to legal reasons.

**Conflicts of Interest:** The authors declare no conflict of interest.

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
