# Peer review of "Machine-Learning-Based Model for Hurricane Storm Surge Forecasting in the Lower Laguna Madre"

_algorithms, doi:10.3390/a16050232_

Round 1
Reviewer 1 Report
The classification and scale of hurricanes may not be significant enough to warrant inclusion as the first paragraph in the introduction. Consideration should be given to the importance of providing a clear and concise introduction that captures the reader's attention and establishes the context and purpose of the paper.
This work is highly interesting because storm surges occur in coastal communities located in the Caribbean, Southeast Asia, and Texas. The paper presents a model that combines Convolutional Neural Networks (CNN) and Long Short-Term Memory (LSTM).
The article should include an explanation of the dataset's characteristics, including the main predictors and data preprocessing. The limitations of the dataset should be discussed in a dedicated section, rather than in the conclusions.
To evaluate the reproducibility and quality of the proposed models, it is important to provide information about the characteristics of the CNN, LSTM, and CNN+LSTM architectures, such as the number of layers, epochs, training time, and other relevant parameters.
Although the model's results are presented, they are not entirely satisfactory. Apart from the data, what other improvements are proposed?"
Author Response
Thank you for your valuable feedback. We have made changes to address your comments and suggestions.
Changes have been made to the introduction section, on page 2, to address the context of the study and reduce the relevance of the hurricane classification scale. More discussion about the data was added in the Modeling scenarios and data processing section, on page 9 at the bottom, to bring up the limitations of the data as well as add to the explanation of its processing (page 7, from line 181 onwards). The forecasting model section was expanded to include details about training parameters and the configuration of the layers in the models as seen on page 10. Further discussion was added at the end of the results section to provide alternatives for improving the model, refer to page 18.
We appreciate your feedback and hope to have addressed all mentioned points,
Best regards,

Reviewer 2 Report
The paper is interesting and is certainly worth publishing, but only after the necessary corrections and additions have been made.
In the literature list, the information on particular items is very incomplete. First of all, for almost all items the year of publication is not given, but the place of publication (journal, conference ?) is also missing. Therefore, it is difficult to assess its relevance and quality.
The forecasting model (page 10) is described insufficiently. It is only known what types of models were used in the study. However, any more detailed information about the architecture of the networks used and their parameters is missing. It is also not described how the combination of CNN and LSTM models was created, with the help of which the third forecasting model was created. It is essential that the above information be completed.
The method of preparing data for the network learning process also needs a more detailed description.
On page 2 (lines 26-30), 2 sentences of text are repeated.
Author Response
Thank you for bringing up the necessary corrections to be made in the draft. We appreciate your feedback and have made changes based on your comments.
More information regarding the year of publication and place has been added to the table of samples of literature in the Materials and Methods section on page 5. In the same manner, more information about the data was added to the Modeling scenarios and data processing section one page 7 from line 181 onwards. Pertinent information about the model’s architecture and training parameters was added on page 10 in the Forecasting model section. Errata on lines 26-30 were removed.
We would like to thank you again for your feedback,
Best regards,

Round 2
Reviewer 1 Report
The authors made the requested modifications. I appreciate their work. The authors present a good technical article with a relevant topic of a very specific problem and an adequate research methodology based on deep learning.
I consider that it is a very interesting paper and matches well to the Algorithms. According to these factors, my recommendation is ACCEPT.
Author Response
We appreciate your time and effort in reviewing and suggesting changes to our manuscript. Thank you.
Best regards,
Cesar Davila
Reviewer 2 Report
The authors have made some corrections and additions to the article, but I still have some objections to it.
1 The list of publications absolutely needs to be improved. Information about the place and year of publication should be included in References, not in the table on page 5. References are still incomplete.
2. In my opinion, there is still no precise information about the architecture of the networks used and their parameters. It is also unclear how the network architecture was selected.
Author Response
Thank you for your valuable feedback on our manuscript. We have made changes to the document as per your suggestions, and we would like to address them here.
Firstly, we have reverted to the original version of table 1 on page 5. We apologize for any confusion that may have been caused. Instead, we have made the necessary changes to the references on the last page of the manuscript as per your recommendation. The revised format now includes the year and place of publication.
We have added lines 282-289 on page 10 to explain how the architecture and parameters were found. We hope this explanation provides adequate insight into how the model progressed iteratively over time as part of an initial pilot study.
Finally, we have also found and corrected some typos and errata that were present in the manuscript.
We appreciate your efforts in reviewing our work, and we hope that these changes meet your expectations. Thank you once again for your valuable feedback.
Sincerely,
Cesar Davila
